# Evaluating the effect of short-course rifapentine-based regimens with or without enhanced behaviour-targeted treatment support on adherence and completion of treatment for latent tuberculosis infection among adults in the UK (RID-TB: Treat): protocol for an open-label, multicentre, randomised controlled trial

Molebogeng X Rangaka,[1,2] Yohhei Hamada [ID],[1] Trinh Duong,[3] Henry Bern,[3] Joanna Calvert,[3] Marie Francis,[1] Amy Louise Clarke,[4] Alex Ghanouni,[4] Charlotte Layton,[3] Vanessa Hack,[1] Ellen Owen-Powell,[3] Julian Surey,[1] Karen Sanders,[3] Helen L Booth,[5] Angela Crook,[3] Chris Griffiths [ID],[6] Robert Horne,[4] Heinke Kunst,[7] Marc Lipman [ID],[8,9] Mike Mandelbaum,[10] Peter J White,[11,12] Dominik Zenner,[1,6] Ibrahim Abubakar [ID][1]

For numbered affiliations see end of article.

**Correspondence to**
Dr Molebogeng X Rangaka;
l.rangaka@ucl.ac.uk

## ABSTRACT

**Introduction** The successful scale-up of a latent tuberculosis (TB) infection testing and treatment programme is essential to achieve TB elimination. However, poor adherence compromises its therapeutic effectiveness. Novel rifapentine-based regimens and treatment support based on behavioural science theory may improve treatment adherence and completion.

**Methods and analysis** A pragmatic multicentre, open-label, randomised controlled trial assessing the effect of novel short-course rifapentine-based regimens for TB prevention and additional theory-based treatment support on treatment adherence against standard-of-care. Participants aged between 16 and 65 who are eligible to start TB preventive therapy will be recruited in England. 920 participants will be randomised to one of six arms with allocation ratio of 5:5:6:6:6:6: daily isoniazid +rifampicin for 3 months (3HR), routine treatment support (control); 3HR, additional treatment support; weekly isoniazid +rifapentine for 3 months (3HP), routine treatment support; weekly 3HP, additional treatment support ; daily isoniazid +rifapentine for 1 month (1HP), routine treatment support; daily 1HP, additional treatment support. Additional treatment support comprises reminders using an electronic pillbox, a short animation, and leaflets based on the perceptions and practicalities approach. The primary outcome is adequate treatment adherence, defined as taking ≥90% of allocated doses within the pre-specified treatment period, measured by electronic

## Strengths and limitations of this study

⇒ The trial allows evaluation of both the effect of two rifapentine-based regimens compared with the standard 3-month daily rifampicin plus isoniazid, and the effect of additional treatment support compared with routine support, on latent tuberculosis infection (LTBI) treatment adherence.

⇒ We will perform process evaluation of the trial interventions, including assessment of intervention acceptability and fidelity, and economic evaluation, which will provide additional evidence to inform treatment options and treatment support.

⇒ The trial is powered to evaluate novel rifapentine-based regimens compared with the standard daily rifampicin plus isoniazid (3HR) and the effect of additional treatment support compared with routine support; however, it does not have sufficient power to evaluate all possible comparisons such as 3-month weekly rifapentine plus isoniazid vs 1-month daily rifapentine plus isoniazid.

⇒ The trial will be conducted in England largely in migrant populations eligible for the LTBI screening programme and contacts of TB patients and thus limiting generalisability to these populations and similar settings.

⇒ Adherence will be measured using electronic pillboxes in all arms while reminders will be activated only in arms with additional treatment support; however, this may impact adherence in control groups.

pillboxes. Secondary outcomes include safety and TB incidence within 12 months. We will conduct process evaluation of the trial interventions and assess intervention acceptability and fidelity and mechanisms for effect and estimate the cost-effectiveness of novel regimens. The protocol was developed with patient and public involvement, which will continue throughout the trial.

**Ethics and dissemination** Ethics approval has been obtained from The National Health Service Health Research Authority (20/LO/1097). All participants will be required to provide written informed consent. We will share the results in peer-reviewed journals.

**Trial registration number** EudraCT 2020-004444-29.

## INTRODUCTION

Successful implementation of screening and treatment for latent tuberculosis infection (LTBI) is critical to further reduce TB incidence globally and achieve TB elimination in low TB incidence countries.[1] A recent call to action issued by the WHO urged for accelerating the scale-up of treatment of LTBI, particularly to mitigate the negative impact from the disruption of TB services caused by the pandemic of COVID-19.[2]

Tuberculosis (TB) in England disproportionately affects underserved communities, such as migrants and homeless people, who consequently experience higher disease burden and worse clinical outcomes. Consequently, in England, LTBI screening and treatment for high risk groups such as new migrants from high TB incidence countries is recognised as an essential strategy to achieve TB elimination.[3] Contact tracing, including testing and treatment of LTBI among contacts, is another essential component of the TB strategy for England.[3]

Achieving optimal treatment adherence and completion is essential to ensure the efficacy of treatment for LTBI and to achieve commensurate reductions in TB incidence. Standard therapeutic options in the UK include 3 months of self-administered daily isoniazid/rifampicin and 6 months of daily isoniazid; the former regimen is often prescribed because of the availability of fixed-dose formulations and its shorter duration. While these regimen are efficacious in preventing disease, their effectiveness is limited by low treatment adherence and completion rates.[4 5] According to data from England in migrants whose treatment outcome is known, 75% completed LTBI treatment between 2019 and 2020.[6 7] The proportion of people who completed treatment varied by Clinical Commission Group (CCG), which was less than 70% in several CCGs.[8]

People with LTBI may need additional support to adhere to effective treatments. Treatment non-adherence can be intentional or unintentional, and is driven by a person's motivation and ability to take medicine as prescribed, respectively.[9] Motivation is influenced by our perceptions (eg, beliefs and preferences) and ability is determined by practical factors (eg, internal capacity and resource).[9] These principles are operationalised as part of the perceptions and practicalities approach to supporting adherence (PAPA) and are applied in National Institute for Health and Care Excellence (NICE) guidelines.[10] The

Necessity and Concerns Framework further explains how patients' motivation to engage with treatment is based on their perceived necessity for, and concerns about, the treatment.[11] Necessity beliefs are influenced by perceptions of the health threat (eg, LTBI) and interpretation of symptoms. The asymptomatic nature of LTBI may negatively impact necessity beliefs, and heighten treatment concerns. As such, intervention to support treatment adherence in people with LTBI will likely be more effective if they address patient beliefs and concerns around treatment, in addition to removing practical barriers.

The need to understand perceptual and practical barriers to treatment adherence, and the potential of advancing technology and drug regimens in the National Health Service (NHS) has been highlighted. Some mobile/digital technology (mHealth) has been shown to improve adherence in TB disease studies. A recent study in China found electronic reminders, using specially designed electronic medication monitors, improved treatment adherence in such TB patients, but multiple two-way daily text messaging reminders, didactic in nature, did not.[12] Most of the evidence available is on TB disease with little research on mHealth interventions to improve LTBI treatment adherence.[13 14] Another call to action issued by the WHO suggested TB preventive treatment programmes should consider communication technologies for medication adherence support.[15] The evidence on mHealth interventions for LTBI treatment would contribute to their global scale-up.

Another approach to promote better treatment adherence and completion is to decrease the complexity of current LTBI regimens. A regimen that is given once weekly may result in better treatment completion than the current daily 3-month regimen. A randomised controlled trial demonstrated that a new regimen of 12 doses of weekly rifapentine and isoniazid (3HP) delivered through direct observation (ie, with patients being supervised taking each dose) is non-inferior to 9 months of daily isoniazid.[16] Our network meta-analysis suggests that 3HP has similar efficacy to the UK standard-of-care of a 12-week, daily isoniazid/rifampicin regimen (3HR).[17] Furthermore, a recent trial in people living with HIV (23% with LTBI as demonstrated by a positive tuberculin skin test and/or Interferon Gamma Release Assay result) demonstrated non-inferiority of daily 1-month rifapentine plus isoniazid (1HP) compared with 9 months of daily isoniazid.[18] The 1-month regimen resulted in better adherence and fewer serious adverse events. Based on this study, and by extrapolating to HIV-negative individuals, newly published WHO guidelines recommend this regimen regardless of HIV status. However, there is no published evaluation of whether these more expensive rifapentine-based regimens lead to better treatment completion than the current daily administered UK standard-of-care. In particular, evidence is limited on the use of 3HP with patient self-administration and no study has compared its completion with 3HR.

To develop tools to reduce TB rates, we need to evaluate advancing technology and drug regimens, but also understand the barriers and enablers of adherence.[3] To date, adherence interventions have predominantly focused on removing practical barriers to adherence (eg, reminder of shortening the drug regimen). However, such approaches applied in isolation ignore patient beliefs. LTBI is asymptomatic which means patients might have a disconnect between medical advice and their perceived need for treatment.[19] NICE guidelines recommend a PAPA to adherence support, whereby beliefs (necessity and concerns) are elicited and addressed in addition to practical barriers.[10 11]

We previously conducted the HALT-LTBI study, a pilot study assessing the safety and treatment completion of 3HP compared with standard care.[20] HALT-LTBI demonstrated the feasibility of recruiting LTBI patients to such a trial; no serious adverse events defined as grade 3 or more were reported, supporting the safety of rifapentine and isoniazid regimens in individuals eligible for LTBI treatment in the UK. 78% and 68% of participants completed treatment in the experimental and standard-of-care arms, respectively, but the pilot was not powered to detect differences in treatment completion. Thus, we will conduct a fully powered trial to compare treatment adherence and adverse events of novel 3HP and 1HP regimens compared with 3HR and to assess the effect of additional treatment support in participants given each regimen.

## Objectives

The primary objective of this trial is to assess the effect of novel rifapentine-based regimens (3HP or 1HP) compared with the standard 90-dose daily rifampicin plus isoniazid (3HR), and the effect of additional treatment support compared with routine support, on LTBI treatment adherence.

The secondary objectives are: (1) to evaluate the effect of LTBI treatment and additional treatment support using alternate measures of adherence outcome; and (2) to compare the frequency of adverse events while on treatment for LTBI, and development of TB within 12 months following treatment. Additionally, we will evaluate the process of delivering the adherence intervention and examine intervention fidelity and acceptability as well as the cost-effectiveness of different treatment options and/or additional treatment support.

## METHOD AND ANALYSIS
### Trial design

A multicentre open-label randomised controlled trial with the following six parallel groups (figure 1):

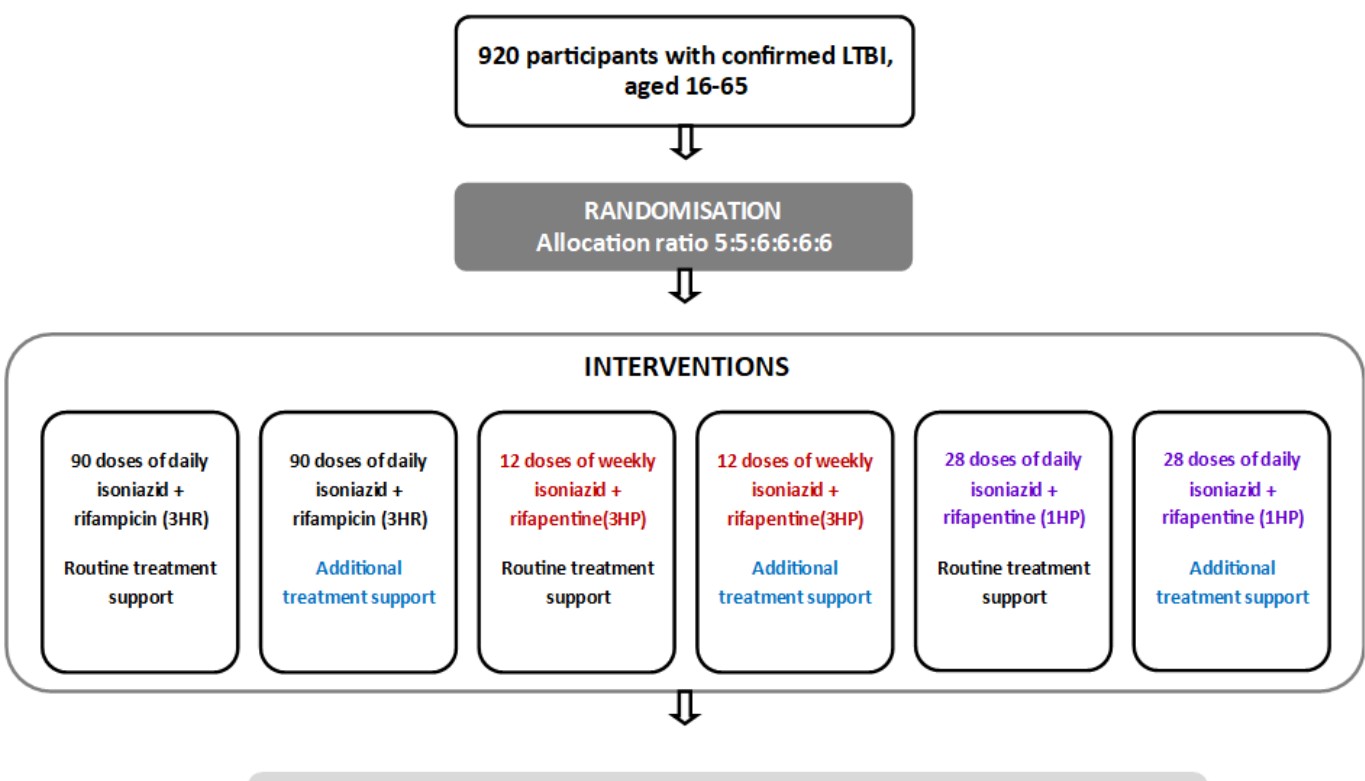

**Figure 1** Trial schema. LTBI, latent tuberculosis infection.

**ARM 1**—Daily isoniazid +rifampicin for 3 months (3HR), routine treatment support (Standard-of-care; control arm)

**ARM 2**—Daily 3HR, additional treatment support.

**ARM 3**—Weekly isoniazid +rifapentine for 3 months (3HP), routine treatment support

**ARM 4**—Weekly 3HP, additional treatment support.

**ARM 5**—Daily isoniazid +rifapentine for 1 month (1HP), routine treatment support.

**ARM 6**—Daily 1HP, additional treatment support

A factorial design was not chosen for several reasons. First, it is anticipated that there will be an interaction between type of regimen and treatment support; additional treatment support is likely to confer a smaller benefit with 3HP/1HP compared with 3HR. Second, the power to detect the effect of an intervention would be reduced if the effect of the second intervention is greater than expected.

### Study setting

The trial will recruit from secondary care sites that provide LTBI treatment in England, UK. RID-TB: Treat is part of a 5-year programme of work (RID-TB) which is funded by the National Institute for Health Research (NIHR) (RP-PG-0217-20009 https://dev.fundinga-wards.nihr.ac.uk/award/RP-PG-0217-20009). We expect to recruit participants from 15 care sites.

### Study population

The trial will enrol populations who are eligible for treatment for LTBI according to the national guidance. We envisage that the majority of individuals eligible for this are contacts of persons diagnosed with TB disease, and/or migrants eligible for the national LTBI screening programme.[21] The LTBI migrant screening programme includes migrants who are aged 16 to 35 years, entered the UK from a high incidence country (≥150/100 000) or sub-Saharan Africa within the last 5 years and had been previously living in that high incidence country for 6 months or longer.[21] Inclusion and exclusion criteria are shown in box 1.

Participants will be identified from secondary care settings in the UK where persons eligible for treatment for LTBI are managed. Participants will be recruited individually, but if any participants share a household, they will be allocated to the same arm as the first person recruited from that household (effectively resulting in randomisation by household).

Non-English speakers will not be excluded from the trial. We will translate patient-facing materials and use interpreters to support non-English-speaking participants.

### Treatment

Participants who are randomised to arms 1 and 2 will receive the standard of care regimen: rifampicin plus isoniazid once daily for 90 doses (3HR).

---

**Box 1  Study inclusion and exclusion criteria**

Inclusion criteria
1. Aged ≥16 years to ≤65 at screening.
2. Latent tuberculosis infection (LTBI) diagnosis defined on the basis of all of the following:
    1. A positive result on an Interferon Gamma Release Assay, Tuberculin Skin Test or C-Tb skin test.
    2. Negative TB symptoms at screening.
    3. No signs of active TB on a chest X-ray.
3. Eligible for LTBI treatment at TB clinics and national LTBI screening services based on National Institute for Health and Care Excellence guidelines, which means having one or more of the following :
⇒ Recent infection (contact tracing).
⇒ New entrants at risk (ie, those that immigrated <5 years from countries with a high incidence of TB, which is defined as ≥40 cases/100 000 population).
⇒ Individuals who are assessed in the TB clinic for latent TB testing, or have been referred for treatment following testing by specialities or departments within primary or secondary care settings.
⇒ Agree to LTBI treatment.
⇒ Willing and able to provide written informed consent.

Exclusion criteria
1. Patients weighing <30 kg.
2. Need for medications that cannot be safely taken together with study drugs (eg, protease inhibitors in people living with HIV and people with refractory epilepsy taking phenytoin/carbamazepine).
3. Any medical condition deserving priority of treatment (such as: porphyria, malabsorption syndromes, *Clostridium difficile*-associated diarrhoea and other conditions).
4. History of sensitivity/intolerance to isoniazid or rifamycins.
5. Individuals with documented liver disease, defined as:
⇒ Liver function tests (alanine aminotransferase/aspartate aminotransferase/bilirubin) over three times upper limit of normal at baseline. This reflects normal clinical practice. For participant safety, liver function tests are carried on a regular basis. One abnormal value prevents the patient from participating on the study.
⇒ Clinical diagnosis of cirrhosis (jaundice, haematemesis, ascites or previous episodes of liver encephalopathy).
⇒ Hepatitis B surface antigen positive or hepatitis C virus antibody positive and deemed ineligible for LTBI treatment by the clinician.
⇒ Intending to move outside of the treatment locality within 20 weeks of starting treatment.
⇒ Individuals who would usually be offered LTBI treatment under Directly Observed Therapy as part of enhanced case management in complex cases such as those from under-served groups (such as people who are homeless, misuse substances, have been in prison or who are vulnerable migrants).
⇒ Use of another experimental investigational medicinal product that is likely to interfere with the study medication within 3 months of study enrolment.
⇒ Women who are breast feeding, pregnant or of childbearing potential who do not agree to use an effective method of contraception from the time consent is signed until 4 weeks after treatment discontinuation or completion. Males whose partners are of childbearing potential must also agree to use an effective method of contraception.
⇒ Women of childbearing potential without a negative urine pregnancy test within 7 days prior to being registered for trial treatment.

---

**Table 1** Doses of study treatment

| | Body weight | | |
|---|---|---|---|
| Arm 1 and 2: rifampicin plus isoniazid once daily for 90 doses (3 months) | <50 kg | ≥50 kg | |
| | 3 x Isoniazid/Rifampicin fixed dose combination (150/100) | 2 x Isoniazid/Rifampicin fixed dose combination (300/150) | |
| Arm 3 and 4: rifapentine plus isoniazid once weekly for 12 doses (3 months) | 30 to <32 KG | 32 to <50 kg | ≥50 kg |
| | Rifapentine 600 mg + Isoniazid 15 mg/kg | Rifapentine 750 mg + Isoniazid 15 mg/kg | Rifapentine 900 mg + Isoniazid 15 mg/kg (900 mg maximum) |
| Arm 5 and 6: rifapentine plus isoniazid once daily for 28 doses (1 month) | 30 to <35 kg | 35 to ≤45 kg | ≥45 kg |
| | Rifapentine 300 mg + 300 mg Isoniazid | Rifapentine 450 mg + 300 mg isoniazid | Rifapentine 600 mg + 300 mg isoniazid |

Participants who are randomised to arms 3 and 4 will receive rifapentine plus isoniazid once weekly for 12 doses (3HP) and those who are randomised to arms 5 and 6 will receive rifapentine plus isoniazid once daily for 28 doses (1HP). Participants will be given a 1-month supply of the medications at every visit in general but it also depends on local practice as this is a pragmatic trial.

In order to account for missed doses and interruption of treatment due to adverse events, participants given 3HR or 3HP will have 16 weeks and those given 1HP will have 6 weeks to complete treatment. In the study by Swindells *et al*, participants were given 8 weeks to complete 1HP.[18] We have chosen 6 weeks to make the period proportionally similar to that for 3HR and 3HP. Clinicians will assess the need for treatment extension based on the assessment of adherence and review of reasons for non-adherence but should not extend beyond recommended grace periods.

In all arms, participants will receive vitamin $B_6$ (pyridoxine). The dosages of study drugs are shown in table 1.

Rifapentine and rifampicin are known to induce the hepatic cytochrome CYP450 enzyme system. Caution is recommended in using medications that are metabolised by this system. Concurrent use of protease inhibitors, hepatitis-C antiviral drugs, or praziquantel is not permitted.

### Treatment support
#### Routine treatment support
Participants allocated to arms 1, 3 and 5 will receive routine treatment support. Participants will be given information about treatment for LTBI including expected adverse events and the importance of adherence, according to local practice. Adherence will be reviewed at each follow-up visit or remote consultation via self-reporting and/or pill count and discussed with the participant. An electronic pill monitor box, Wisepill EvriMed1000 (Wisepill, Somerset West, South Africa)[22] will collect the date and time of each opening to collect information on adherence. However, it will be set to silent mode and not be used as an adherence reminder tool.

### Intervention
Participants assigned to arms 2, 4 and 6 will receive a PAPA-based intervention designed to provide additional treatment support (ie, in addition to routine treatment support).[11] Specifically, the intervention will consist of an animation which will (1) provide a rationale for treatment necessity and help people understand how LTBI treatment can help them to achieve a health goal that is important to them, (2) address common concerns about LTBI treatment and (3) address practical barriers to treatment (eg, anchoring treatment to daily activities). The animation will be supported by a leaflet that covers misperceptions about LTBI testing and treatment, and other frequently asked questions. Participants will also be asked to set reminders using an electronic pill monitor box (Wisepill EvriMed). The electronic pillbox allows two modes of reminders: audio alarm from the box or text-message to participants' mobile phones.

The reminder can be set at prespecified times and can also be activated to send a reminder when the pill box is not opened. Site staff will discuss options with each participant and set reminders according to their preferences. Participants can opt not to receive reminders before or at the time of intended medication intake. However, they will still be reminded when the box is not opened within a pre-specified time in a day and they will receive a supportive text message automatically sent by the pillbox. The mode of reminder can be further adjusted during the course of treatment as necessary on discussion with a clinician. The pillbox will electronically collect the date and time of each opening.

## Study assessment and follow-up
### Screening, randomisation and baseline assessment
Randomisation and baseline assessment will occur on the same day (week 0). In some cases, this may also be the same day as Screening. Following informed consent procedures, participants will be screened for eligibility. A TB symptom screen and urine pregnancy test will be carried out, and data on the participant's TB risk group category will be collected. Demographic and medical history information will be collected. We will check the results of clinical, laboratory and radiological assessments performed under routine care before entry to the trial to confirm eligibility. A TB symptom screen and urine pregnancy test will be repeated at the randomisation/baseline visits unless the screening and randomisation visits occur on the same day.

### Assessment of adherence
Assessment of adherence will be primarily measured using the Wisepill, which collects the date and time of each opening. Adherence will also be measured through self-reporting and pill count under routine care either at physical clinic visits or remote consultations as per the local standard. Attending clinicians will count the number of remaining tablets. The difference between the number of tablets dispensed and the number returned will be calculated.

### Clinical assessment during follow-up
As per usual practice, liver function tests (hepatic transaminases, alanine aminotransferase/aspartate aminotransferase and total bilirubin) will be performed at week 2 for all participants. Afterwards, liver function tests will be performed at weeks 4, 8, 12 and 16 while on treatment and at completion, or at other times if deemed necessary by attending clinicians (eg, abnormality in preceding tests, new onset of symptoms suggesting potential liver toxicity). These tests should be performed at any time during the treatment and post-treatment phase if the participant exhibits symptoms or signs of drug-induced liver injury.

Adverse events expected with study drugs will be clinically assessed at every visit. These include anorexia, nausea, vomiting, fatigue, weakness, jaundice, rash, peripheral neuropathy and bruising. Participants who already completed treatment and have no scheduled visits will be given a phone call at week 8, 12, 16 and 20 to check adverse events and TB signs and symptoms since the last dose.

At every physical visit or remote consultation, symptoms and signs of TB disease will be reviewed as well as concomitant medications using a brief questionnaire. There will be no formal study visits after completion of treatment.

## Protocol treatment discontinuation
An individual participant may stop treatment early or trial participation be stopped early for any of the following reasons: Unacceptable toxicity or adverse event including (eg, serious adverse events leading to discontinuation of treatment); intercurrent illness that prevents further treatment; active TB disease; any change in the participant's condition that justifies the discontinuation of treatment in the clinician's opinion; pregnancy; inadequate compliance with the protocol treatment that preclude treatment within allowable time-frame in the judgement of the treating physician; and withdrawal of consent for treatment by the participant.

## Outcomes
### Primary outcome
The primary outcome is adequate treatment adherence, defined as taking ≥90% of allocated doses within the allowable time frame from randomisation (binary outcome). For the primary analyses, treatment adherence is measured using an electronic monitor box

### Secondary outcomes
The secondary outcome measures are:
► Effectiveness: (1) proportion of allocated doses missed over the treatment period (measured using monitor box); (2) proportion of allocated pills missed over the treatment period (measured using pill counts); (3) taking at least 90% of doses and pills over the treatment period (binary outcome assessed using both monitor box and pill counts) and (4) early study treatment discontinuation for any reason
► Safety: (1) permanently stop study treatment due to drug-related adverse events (ie, adverse reactions); (2) experience Grade ≥3 adverse events and (3) develop TB disease within 12 months.

## Sample size
The six-arm design allows evaluation of:
► The effect of the novel treatment regimens (3HP and 1HP) vs standard-of-care regimen (3HR), under routine treatment support.
► The effect of additional treatment support vs routine treatment support for each individual regimen.

A total of 920 participants are to be recruited. This provides 80% power for each of the following comparisons:
► Arm 3 vs Arm 1—that is, 3HP+routine treatment support vs 3HR+routine treatment support.
► Arm 5 vs Arm 1—that is, 1HP+routine treatment support vs 3 HR+routine treatment support.
► Arm 2 vs Arm 1—that is, 3HR+additional treatment support vs 3HR+routine treatment support.
► Arm 4 vs Arm 3—that is, 3HP+additional treatment support vs 3HP+routine treatment support.
► Arm 6 vs Arm 5—that is, 1HP+additional treatment support vs 1HP+routine treatment support.

The power calculations assume the following:
► 70% adherence rate in Arm 1.
► 3HP and 1HP improve adherence rate by 15% (absolute difference) compared with 3HR, respectively, with routine treatment support.[18 23 24]

- ► Compared with routine treatment support, additional treatment support improves adherence rate by 15% for 3HR, and 10% for 3HP and 1HP, respectively.[12]
- ► Two-sided alpha 5% (see below for type I error considerations).
- ► Average number of participants enrolled per household is 2, taking into account the average household size in UK.[25]
- ► Intra-class correlation within a household is 0.1

The 70% adherence rate assumed for Arm 1 is based on the 77% LTBI treatment completion rate reported from the Public Health England LTBI testing and treatment database for 2018.[26]

### Randomisation and allocation

Participants will be randomised centrally using a computerised algorithm developed and maintained by the Medical Research Council Clinical Trials Unit at University College London (MRC-CTU). To randomise a participant, the information contained on a completed Randomisation Form will be entered into the secure online trial database by trial team members at the site who have been trained and authorised to randomise by the MRC-CTU. The database will automatically check for eligibility. Only those who meet all eligibility criteria will be able to be randomised. Randomisation will be performed using minimisation with an additional random element, to be balanced with respect to centre and TB exposure risk group.

### Blinding

This is an open-label trial. Blinding of participants and care providers to the allocation group is not relevant since the primary objective of this trial is to examine the effect of shorter or weekly regimens and additional treatment support on treatment adherence.

### Data collection methods and management

Adherence data will be collected through the Wisepill monitor box. Demographic and clinical information will be collected through clinical consultation and recorded on relevant worksheets. Development of TB within 12 months after starting treatment and outcomes of pregnancy that are found after enrolment will be collected using records held by NHS Digital, Public Health England and/or the National TB register.

The trial will be conducted in compliance with the UK Data Protection Act 2018 (DPA number: Z6364106) and the EU Regulation General Data Protection Regulations 2016/679/EC (GDPR) for protection of personal data.

### Statistical methods

The estimands for the primary analyses are defined in table 2. The primary analyses will compare the proportion of participants with adequate adherence between arms using the following approach: (a) Arm 3 vs Arm 1—that is, 3HP+routine treatment support vs 3HR+routine treatment support; (b) Arm 5 vs Arm 1—that is, 1HP+routine treatment support vs 3HR+routine treatment support and (c) Arm 2 vs Arm 1—that is, 3HR+additional treatment support vs 3HR+routine treatment support

If comparison (a) shows 3HP improves adherence compared with 3HR, then additional treatment support will be formally tested for 3HP by comparing Arm 4 vs Arm 3—that is, 3HP+additional treatment support vs 3HP+routine treatment support; otherwise, the adherence rates will be compared between these arms as

**Table 2** Definition of the estimands for the primary analyses

| Attribute | Definition |
|---|---|
| Treatments | The primary analyses are based on the following comparisons: (a) Arm 3 vs Arm 1—that is, 3HP+routine treatment support vs 3HR+routine treatment support (b) Arm 5 vs Arm 1—that is, 1HP+routine treatment support vs 3HR+routine treatment support (c) Arm 2 vs Arm 1—that is, 3HR+additional treatment support vs 3HR+routine treatment support<br>If comparison (a) shows 3HP improves adherence compared with 3HR, then additional treatment support will be formally tested for 3HP by comparing Arm 4 vs Arm 3—that is, 3HP+additional treatment support vs 3HP+routine treatment support. Additional treatment support will be similarly assessed for 1HP. |
| Population | Adults aged 16–65 years diagnosed with LTBI and eligible for LTBI treatment. |
| Endpoint | Adequate treatment adherence, defined as taking ≥90% of allocated doses within the allowable time frame. |
| Intercurrent events | The main intercurrent events and how they will be handled in the estimand are as follows:<br>► Failure to collect all prescriptions—composite and treatment policy strategies lead to same estimated effect.<br>► Early treatment discontinuation for any reason including adverse event(s) and active TB: a treatment policy strategy will be used, that is, the participant is considered to have stopped treatment regardless of the occurrence of the intercurrent event. |
| Population-level summary measure | Risk ratio for adequate treatment adherence comparing the relevant arms. |

LTBI, latent tuberculosis infection.

exploratory analyses. Additional treatment support will be similarly assessed for 1HP.

All randomised patients will be included in the primary analyses, apart from those subsequently found to have had TB disease at baseline but enrolled in error (modified intention-to-treat approach). The risk ratio (with 95% CI) for adequate treatment adherence comparing the relevant arms will be estimated using log-binomial generalised linear mixed models, allowing for intra-household correlation.

Type I error adjustment for multiple comparisons is not deemed necessary since:

► The research hypotheses corresponding to comparisons (a), (b) and (c) are considered sufficiently distinct.[27–29]
► The effect of additional treatment support vs routine support is being evaluated in non-overlapping populations for 3HR, 3HP and 1HP, respectively.
► The closed test approach whereby the effect of additional treatment support will only be formally tested for 3HP if there is evidence that 3HP improves adherence compared with 3HR with routine treatment support protects the type I error. This approach will also be used for the assessment of additional treatment support for 1HP.

For participants who have collected all prescriptions but are lost to follow-up before completing treatment, the adherence data until the end of allocated period can still be downloaded remotely from the Wisepill monitor box to ascertain whether adequate treatment adherence

is achieved; these data will be included in the primary analyses. In sensitivity analyses, the primary outcome will be imputed for these patients using multiple imputation by chained equations, with imputation to be conducted separately by study arm. Sensitivity analyses will also be performed assuming no drug intake from the last follow-up visit attended.

Supplementary analyses will consider different definitions of adequate treatment by varying the minimum proportion of doses required to have been taken, and different allowable time frames for making up missed doses. In addition, other analysis populations will be considered, including intention-to-treat and per protocol (including only participants who commenced their original allocated trial intervention). Planned exploratory subgroup analyses, will examine outcomes in predefined subgroups.

### Safety reporting

The definitions of the EU Directive 2001/20/EC Article 2 based on the principles of Good Clinical Practice apply to this trial protocol. These definitions are given in table 3. All grade 3 or higher adverse events, whether expected or not, will be recorded in the patient's medical notes. All adverse events will be recorded up to week 20. Serious adverse events should be notified to the CTU within 24 hours of the investigator becoming aware of the event from the time of randomisation to the last assessment of adverse events, that is, week 20. Adverse events will be graded using the Division of AIDS toxicity grading scale.[30]

**Table 3** Definitions of adverse events (AE) and reactions

| Term | Definition |
|---|---|
| AE | Any untoward medical occurrence in a patient or clinical trial participant to whom a medicinal product has been administered including occurrences that are not necessarily caused by or related to that product. |
| Adverse reaction (AR) | Any untoward and unintended response to an investigational medicinal product related to any dose administered. |
| Unexpected AR | An AR, the nature or severity of which is not consistent with the information about the medicinal product in question set out in approved Reference Safety Information for that product in the trial. |
| Serious AE (SAE) or serious AR or suspected unexpected serious AR | Any AE, AR or unexpected AR that:<br>► Results in death<br>► Is life-threatening*<br>► Requires hospitalisation or prolongation of existing hospitalisation†<br>► Results in persistent or significant disability or incapacity<br>► Consists of a congenital anomaly or birth defect<br>► Is another important medical condition‡ |

*The term life-threatening in the definition of a serious event refers to an event in which the patient is at risk of death at the time of the event; it does not refer to an event that hypothetically might cause death if it were more severe, for example, a silent myocardial infarction.
†Hospitalisation is defined as an inpatient admission, regardless of length of stay, even if the hospitalisation is a precautionary measure for continued observation. Hospitalisations for a pre-existing condition, that has not worsened or for an elective procedure do not constitute an SAE.
‡Medical judgement should be exercised in deciding whether an AE or AR is serious in other situations. The following should also be considered serious: important AEs or ARs that are not immediately life-threatening or do not result in death or hospitalisation but may jeopardise the subject or may require intervention to prevent one of the other outcomes listed in the definition above; for example, a secondary malignancy, an allergic bronchospasm requiring intensive emergency treatment, seizures or blood dyscrasias that do not result in hospitalisation or development of drug dependency.

Participants may be able to claim compensation for injury caused by their participation in the clinical trial in accordance with the insurance policy held at UCL.

## Monitoring and oversight

The trial will be monitored by the MRC-CTU. An Independent Data Monitoring Committee (IDMC) will be formed. The IDMC will review study conduct and safety data regularly. The IDMC will be asked to advise on whether the accumulated data from the trial, together with results from other relevant trials, justify continuing recruitment of further participants. The IDMC will make recommendations to the Programme Steering Committee (PSC) as to whether the trial should continue in its present form.

The PSC has membership from the Trial Management Group plus independent members (approved by NIHR), including the chair and patient and public involvement (PPI) contributors. The role of the PSC is to provide overall supervision for the trial and provide advice through its independent chair. The ultimate decision for the continuation of the trial lies with the PSC.

## Process evaluation

The process evaluation will follow MRC guidance using an embedded, mixed-methods evaluation approach in order to assess acceptability, fidelity, and mechanisms of effects of the interventions. It will be conducted by the research team, working closely with the Intervention Development Group and clinicians delivering the trial.

### Patient sample

Patients in the full trial sample will be administered validated questionnaires assessing the psychological characteristics that we predict will mediate the effects of the interventions. Questionnaires will be administered during scheduled clinic appointments at baseline (0 weeks), interim (2 weeks) and treatment completion (either 4 or 12 weeks depending of regimen). Baseline measure will include Beliefs about Medicines Questionnaire (BMQ-Specific/BMQ-General), Perceived Sensitivity to Medicines Scale-5, Brief illness perceptions questionnaire (BIPQ), The Satisfaction with Information about Medicines Scale, Hospital Anxiety and Depression Scale. At follow-up, participants will complete the BIPQ and BMQ-Specific, and a measure of self-reported adherence (Medication Adherence Report–5) and the Treatment intrusiveness Questionnaire. A subset of participants will also be approached for a qualitative assessment of their experiences in the trial. Participants in each intervention arm will be purposively sampled based on their treatment adherence (10 participants per arm: 5 high adherence, 5 low adherence; total 60 interviews; adherence in line with the primary outcome). Measures will consist of brief, semi-structured interviews.

### Staff sample

Healthcare professionals responsible for administering the interventions will be requested to complete a short checklist form following patient randomisation in order to assess intervention fidelity. This will confirm whether each component of the interventions was delivered per protocol. We will also purposively sample 20 service providers to take part in brief, semi-structured interviews (in person or by phone) in order to obtain feedback on the delivery of the intervention and to identify any issues that might enhance delivery in practice. In addition, we will use these interviews to investigate wider contextual issues impacting on delivery. We will also encourage implementing clinicians to report major issues that might compromise intervention delivery during the trial, rather than waiting for a formal interview on trial completion.

## Health economic evaluation

This will estimate if changes to LTBI diagnosis and/or treatment are cost-effective from the perspective of the NHS, using a health-economic model to synthesise data obtained within the entire RID-TB programme and evidence from other sources. Participants will be asked to complete monthly EQ-5D questionnaires. We will collect information on the costs participants incur in attending appointments within this trial, to allow potential future analysis from a societal perspective.

## PATIENT AND PUBLIC INVOLVEMENT

The trial was discussed with the charity TB Alert and two community representatives drawn from a migrant charity and a patient previously treated for LTBI. A charity representative and one former patient read versions of the grant proposal and contributed suggestions on study design. At the protocol development stage, the following input was sought from TB Alert: study design, treatment support interventions, Participant Information Sheet and Consent form, patient-facing questionnaires used for behavioural studies.

During the trial, we will engage with (1) The RID-TB PPI Advisory Group consisting of members recruited via social media accounts, TB nurses, TB patient advocates, ex-patient contacts and voluntary/community organisations and[2] The TB Action Group (TAG) network of people personally affected by TB. We will seek input for: recruitment, patient/public engagement tools, provision of translated materials on LTBI and access to recruitment sites.

## Dissemination

We will report findings of the trial through publications in national and international conferences as well as in peer-reviewed journals. We will follow publication policies used for clinical trials coordinated by the MRC CTU. All headline authors in any publication arising from the main study or substudies must have made a substantive academic or project management contribution to the work that is being presented. Findings will be also disseminated via TB Alert, Treatment Action Group, social media and institutional websites.

## Trial status

The trial has not yet started recruitment. We expect to start recruitment on 1 September 2022 and the trial will close when all participants have completed follow-up (ie, 12 months after initiation of treatment), record linkage to ascertain TB has been finished, and after the trial database is locked, which is anticipated to be within 3 months after information on primary and secondary outcomes have been collected.

## Protocol version and date

This protocol is an abbreviated version of the protocol V.3.0, October 2020.

**Author affiliations**

¹Institute for Global Health, University College London, London, UK
²School of Public Health, and Clinical Infectious Disease Research Institute-AFRICA, University of Cape Town, Cape Town, South Africa
³MRC Clinical Trials Unit at UCL, Institute of Clinical Trials and Methodology, London, UK
⁴Centre for Behavioural Medicine, UCL School of Pharmacy, London, UK
⁵North Central London Tuberculosis Service, Whittington Health NHS Trust and University College London Hospitals NHS Foundation Trust, London, UK
⁶Wolfson Institute for Population Health Barts and the London School of Medicine and Dentistry, Queen Mary University, London, UK
⁷Blizard Institute, Barts and The London School of Medicine and Dentistry, Queen Mary University, London, UK
⁸UCL Respiratory, Division of Medicine, University College, London, UK
⁹Royal Free London Hospital NHS Foundation Trust, London, UK
¹⁰TB Alert, Brighton, UK
¹¹Modelling and Economics Unit, National Infection Service, Public Health England, London, UK
¹²MRC Centre for Global Infectious Disease Analysis, Imperial College, London, UK

**Acknowledgements** We thank the NIHR programme officers, UCL-NIHR Patient and Public Involvement Advisory Group, MRC CTU at UCL Protocol Review Committee and the independent Programme Steering Committee for their support and inputs during the development of the protocol.

**Contributors** IA and MXR conceived the study. IA and MXR led the application to secure funding. IA, MXR, TD, YH, HB, JC, MF, ALC, AG, VH, EO-P, JS, KS, HLB, AC, CG, RH, MJ, HK, ML, MM, PJW and DZ contributed to developing the study design. CL, TD and AC provided statistical oversight. MXR, TD and YH drafted and revised the manuscript. All authors contributed critical intellectual input and approved the final manuscript.

**Funding** The protocol reported in this publication was supported by the NIHR (RP-PG-0217-20009) and will receive support from the NIHR Clinical Research Network (NIHR CRN). Additional funding for patient and public involvement was provided by the UCLH/UCL Biomedical Research Centre Patient & Public Involvement bursary fund. ALC and TD are supported by MRC Grant: MC_UU_12023/27.

**Competing interests** None declared.

**Patient consent for publication** Not applicable.

**Provenance and peer review** Not commissioned; externally peer reviewed.

**ORCID iDs**

Yohhei Hamada http://orcid.org/0000-0002-9845-4267
Chris Griffiths http://orcid.org/0000-0001-7935-8694
Marc Lipman http://orcid.org/0000-0001-7501-4448
Ibrahim Abubakar http://orcid.org/0000-0002-0370-1430

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
