## [Reviewer comments · BMJ Open]

ARTICLE DETAILS

TITLE (PROVISIONAL)	Evaluating the effect of short-course rifapentine-based regimens with or without enhanced behaviour-targeted treatment support on adherence and completion of treatment for latent tuberculosis infection among adults in the UK (RID-TB: Treat): protocol for an open-label, multicentre, randomised controlled trial
AUTHORS	Hamada, Yohhei; Rangaka, Molebogeng; Duong, Trinh; Bern, Henry; Calvert, Joanna; Francis, Marie; Clarke, Amy; Ghanouni, Alex; Layton, Charlotte; Hack, Vanessa; Owen-Powell, Ellen; Surey, Julian; Sanders, Karen; Booth, Helen; Crook, Angela; Griffiths, Chris; Horne, Robert; Kunst, Heinke; Lipman, Marc; Mandelbaum, Mike; White, Peter; Zenner, Dominik; Abubakar, Ibrahim

VERSION 1 – REVIEW

REVIEWER	Shibu, Vijayan
REVIEW RETURNED	01-Dec-2021

GENERAL COMMENTS	There are multiple interventions planned in the intervention arms, however it is not clear how the study team is going to decipher the co-relates of adherence. How the adherence is measured when a missed dose episode is intervened by health worker and brought back the patient on track , where do you attribute the contributor of adherence, technology flags the episode of non opening, but the health worker completes the loop in the above case. Hence it is suggested to address this and resubmit.
---

REVIEWER	Nyang'wa, Bern-Thomas London School of Hygiene & Tropical Medicine
REVIEW RETURNED	11-Dec-2021

GENERAL COMMENTS	Thank you for sharing this well designed and comprehensive study protocol aimed at primarily answering the important question of the role of each LTBI treatment regimen and adherence intervention package, as evidence already exists on the overall safety and efficacy non-inferiority of the newer regimens. I was not able to understand from the manuscript whether study participants who don't speak, understand or write English adequately will be excluded or other steps will be taken and if those measures will then be taken into consideration in the analysis. This of course is pertinent given the expected population to require LTBI treatment may consist of a non negligible proportion whom English may not be their first language.
---

	Can the authors please clarify at what regularity will the participants for each arm receive a prescription and for what duration will the prescription be? Given one extreme of poor adherence is early discontinuation, it is unclear from the manuscript whether those who discontinue treatment will be considered to have missed pills up to the total of the current visit prescription or up to the prescription total of the treatment arm (i.e. total pills for the whole treatment duration). Finally, the trial registry indicates that the trial received ethics approval at the end of 2019, authors should indicate in the manuscript whether the study has started or not and when it is expected to be completed.
--	---

VERSION 1 – AUTHOR RESPONSE

Reviewer: 1

Dr. Vijayan Shibu

Comments to the Author:

There are multiple interventions planned in the intervention arms, however it is not clear how the study team is going to decipher the co-relates of adherence. How the adherence is measured when a missed dose episode is intervened by health worker and brought back the patient on track , where do you attribute the contributor of adherence , technology flags the episode of non opening , but the health worker completes the loop in the above case . Hence it is suggested to address this and resubmit .

Response: As noted in the statistical method section, our trial will allow assessing the impact of additional treatment support including electronic pillboxes compared to the standard of care alone, in which participants will be intervened by health workers if necessary as per the routine care. Therefore, should we find improvement in the adherence in the intervention arms, we can attribute that to the trial interventions. Indeed, since additional treatment support is a package comprising multiple interventions, the main analysis will not be able to pinpoint what intervention actually worked. However, the process evaluation described in the protocol will give some insights into the mechanisms of effects of the interventions

Reviewer: 2

Dr. Bern-Thomas Nyang'wa, London School of Hygiene & Tropical Medicine

Comments to the Author:

Dear Authors,

Thank you for sharing this well designed and comprehensive study protocol aimed at primarily answering the important question of the role of each LTBI treatment regimen and adherence intervention package, as evidence already exists on the overall safety and efficacy non-inferiority of the newer regimens.

I was not able to understand from the manuscript whether study participants who don't speak, understand or write English adequately will be excluded or other steps will be taken and if those measures will then be taken into consideration in the analysis. This of course is pertinent given the expected population to require LTBI treatment may consist of a non negligible proportion whom English may not be their first language.

Response: Non-English speakers will not be excluded from the trial. We will translate patient-facing materials and use interpreters to support non-English speaking participants. The same information has been added to the text of the manuscript.

Can the authors please clarify at what regularity will the participants for each arm receive a prescription and for what duration will the prescription be? Given one extreme of poor adherence is early discontinuation, it is unclear from the manuscript whether those who discontinue treatment will be considered to have missed pills up to the total of the current visit prescription or up to the prescription total of the treatment arm (i.e. total pills for the whole treatment duration).

Response: We have updated the statistical method section to clarify for participants who have collected all prescriptions but are lost to follow-up before completing treatment, the adherence data until the end of the allocated period can still be downloaded remotely from the Wisepill monitor box to measure the number of doses taken and ascertain whether adequate treatment adherence is achieved.

Participants will be given a 1-month supply of the medications at every visit in general but it also depends on local practice as a pragmatic trial.

Finally, the trial registry indicates that the trial received ethics approval at the end of 2019, authors should indicate in the manuscript whether the study has started or not and when it is expected to be completed.

Response: We have added the trial status as follows:

“The trial has not yet started recruitment. We expect to start recruitment on 1 September 2022 and the trial will close when all participants have completed follow-up (i.e. 12 months after initiation of treatment), record linkage to ascertain TB has been finished, and after the trial database is locked, which is anticipated to be within 3 months after information on primary and secondary outcomes have been collected.”

VERSION 2 – REVIEW

REVIEWER	Nyang'wa, Bern-Thomas London School of Hygiene & Tropical Medicine
REVIEW RETURNED	22-Jan-2022

GENERAL COMMENTS	Thank you for making the edits and clarifications.
--

REVIEWER	
REVIEW RETURNED	

GENERAL COMMENTS	
--

REVIEWER	
REVIEW RETURNED